# Mitochondria in Multi-Directional Differentiation of Dental-Derived Mesenchymal Stem Cells

**DOI:** 10.3390/biom14010012

**Published:** 2023-12-21

**Authors:** Haotian Liu, Ke Xu, Yifan He, Fang Huang

**Affiliations:** Hospital of Stomatology, Guanghua School of Stomatology, Sun Yat-sen University, Guangdong Provincial Key Laboratory of Stomatology, Guangzhou 510000, China; liuht35@mail2.sysu.edu.cn (H.L.); xuke28@mail2.sysu.edu.cn (K.X.)

**Keywords:** dental-derived mesenchymal stem cells, multi-directional differentiation, mitochondrial energy metabolism, mitochondrial dynamics, mitophagy, mitochondrial biogenesis

## Abstract

The pursuit of tissue regeneration has fueled decades of research in regenerative medicine. Among the numerous types of mesenchymal stem cells (MSCs), dental-derived mesenchymal stem cells (DMSCs) have recently emerged as a particularly promising candidate for tissue repair and regeneration. In recent years, evidence has highlighted the pivotal role of mitochondria in directing and orchestrating the differentiation processes of DMSCs. Beyond mitochondrial energy metabolism, the multifaceted functions of mitochondria are governed by the mitochondrial quality control (MQC) system, encompassing biogenesis, autophagy, and dynamics. Notably, mitochondrial energy metabolism not only governs the decision to differentiate but also exerts a substantial influence on the determination of differentiation directions. Furthermore, the MQC system exerts a nuanced impact on the differentiation of DMSCs by finely regulating the quality and mass of mitochondria. The review aims to provide a comprehensive overview of the regulatory mechanisms governing the multi-directional differentiation of DMSCs, mediated by both mitochondrial energy metabolism and the MQC system. We also focus on a new idea based on the analysis of data from many research groups never considered before, namely, DMSC-based regenerative medicine applications.

## 1. Introduction

Regenerative medicine has been a focus of research for decades due to the desire to repair and regenerate damaged tissue. Dental-derived mesenchymal stem cells (DMSCs) have recently emerged as promising seed cells for tissue repair and regeneration among the various types of mesenchymal stem cells (MSCs). In recent years, the study of trans-differentiation has gained significant attention, particularly with respect to DMSCs. The aim of this review is to provide an in-depth analysis of the current state of the art in improving and regulating the multilineage differentiation of DMSCs. Recent studies underscore the critical involvement of mitochondria, often referred to as the powerhouses of cells, in steering the differentiation processes of DMSCs. For instance, our latest research has demonstrated that melatonin facilitates odontoblastic differentiation of DMSCs via malic enzyme 2-mediated mitochondrial fusion and respiration [1].

Specifically, this review delves into the latest advancements in understanding how mitochondrial energy metabolism and the mitochondrial quality control (MQC) system, namely biogenesis, autophagy, and dynamics, orchestrate the direction and progression of DMSC differentiation. Processes such as oxidative phosphorylation (OXPHOS), glycolysis, energy metabolism process conversion, reactive oxygen species (ROS), and mitochondrial energy metabolism collectively exert an indispensable influence on DMSC differentiation. Key factors in the MQC system regulate the quality and mass of mitochondria, promote mitochondrial homeostasis, and affect DMSC differentiation. With a more profound comprehension of the complex interaction between mitochondria and DMSCs, there exists significant potential for the creation of innovative and efficacious therapies that exploit the potential of DMSCs to enhance patient outcomes, such as tissue regeneration, functional reconstruction, and cell therapy. Nevertheless, several key questions remain unanswered regarding the specific mechanisms underlying mitochondrial regulation of DMSC differentiation.

## 2. Dental-Derived Mesenchymal Stem Cells

MSCs, originally reported by Friedenstein et al. in the 1970s [2,3] and later termed by Caplan in 1991 [4], have garnered significant attention for their remarkable multipotency and capacity to differentiate into mesodermal lineages [5]. Over the years, MSCs have been detected in a multitude of tissues and organs, including but not restricted to, bone marrow [2], heart [6], skeletal muscle [7], adipose tissue [8], and skin [9], among others. MSCs are crucial for maintaining tissue homeostasis and have emerged as promising candidate cells for regenerative medicine and tissue engineering applications.

DMSCs represent a subpopulation of adult stem cells sourced from dental tissues, with the first successful isolation from adult dental pulp reported in 2000, known as dental pulp stem cells (DPSCs) [10]. Since then, DMSCs have been isolated and characterized from various tissues in the oral cavity, including stem cells from exfoliated deciduous teeth (SHED) [11], periodontal ligament stem cells (PDLSCs) [12], gingival mesenchymal stem cells (GMSCs) [13], dental follicle stem cells (DFSCs) [14], dental papilla cells (DPCs) [15] and stem cells from apical papilla (SCAPs) [16]. Similar to other MSCs, DMSCs exhibit self-renewal ability, immunomodulatory properties, and multiple differentiation potential, including odontoblastic differentiation, osteoblastic differentiation, neurogenic differentiation, angiogenic differentiation, myogenic differentiation, and adipogenic differentiation (Figure 1) [17]. In addition, DMSCs have significant advantages over other MSCs, primarily due to the accessibility and tractability of dental tissues, as well as their ability to serve as a model system for studying their function and properties in vivo. Notably, while DMSCs may express the same typical mesenchymal markers, there are still differences among them. For instance, SCAPs exhibit a prominently higher proliferative rate and a superior mineralization ability compared with DPSCs, while GMSCs can be isolated from gingival tissue in biopsy without compromising tooth integrity [18]. Overall, due to their ease of access, remarkable capacity for vitro expansion, differentiation potential, and lack of complex ethical issues, DMSCs are regarded as ideal candidates for tissue repair and regenerative medicine, providing a practical and attractive source for cell-based regenerative therapies with significant clinical potential. Consequently, enhancing the functions of DMSCs and directing their differentiation are vital steps toward optimizing their use in clinical applications.

## 3. Mitochondria

Mitochondria, the dynamic organelles with double membranes found in eukaryotic cells, are pivotal for energy production and play essential roles in various biological processes (Figure 2). Additionally, studies have suggested that mitochondria operate as a collective, with their activity being tightly orchestrated by the MQC system [19]. This system comprises three crucial processes, namely mitochondrial dynamics, mitophagy, and mitochondrial biogenesis. In recent years, our understanding of mitochondrial dynamics has witnessed a paradigm shift, with the discovery of intercellular mitochondrial transfer. This process represents an extension of both intracellular mitochondrial movement and intercellular communication.

While it was traditionally assumed that changes in energy metabolism simply accompany DMSCs’ multi-directional differentiation to support the differential metabolic demands, numerous studies over the past few decades have demonstrated that changes in energy metabolism can actually dictate the fate of DMSCs, with the mitochondrial quality control system playing a crucial role in regulating DMSCs multi-directional differentiation [20,21].

### 3.1. Mitochondrial Energy Metabolism

The mitochondria play a critical role in cellular energy metabolism, primarily through the main pathways of glycolysis, OXPHOS, and the tricarboxylic acid (TCA) cycle. Under aerobic conditions, pyruvate produced by glycolysis enters the mitochondrial matrix to participate in the TCA cycle, while in hypoxic environments, cells switch to anaerobic respiration, where pyruvate is eventually reduced to fermentation products in the cytoplasm. Furthermore, glycolysis and the TCA cycle produce energetic molecules that are utilized by OXPHOS to synthesize ATP and reduce O2. Moreover, mitochondria are the primary source of intracellular ROS, generating substantial amounts of ROS through the respiratory chain or as byproducts of OXPHOS [22]. During oxidative stress in cells, the dramatically increased ROS can cause cell dysfunction and tissue death, which can severely damage cells. In the context of normal respiration, it is noteworthy that healthy mitochondria consistently produce low levels of superoxide. The presence of various endogenous scavengers is imperative for the effective mitigation of excessive ROS. Specifically, manganese superoxide dismutase (MnSOD), situated within the mitochondrial matrix, expeditiously catalyzes the conversion of superoxide to hydrogen peroxide, a reactive molecular species [23]. Subsequently, this hydrogen peroxide undergoes transformation into water by either catalase or glutathione peroxidase within the mitochondria or following diffusion in the cytosol. Beyond the realm of enzymes, cellular defense extends to a repertoire of antioxidant molecules, exemplified by glutathione, ascorbic acid, and α-tocopherol assuming roles of quenching ROS [24].

### 3.2. Mitochondrial Dynamics

Mitochondrial dynamics primarily encompass mitochondrial fusion and fission, two opposing processes that act synergistically to maintain appropriate morphology, size, mass of mitochondria, and their physiological function. As mitochondria are organelles with double membranes, mitochondrial fusion involves two membrane fusion events: outer membrane fusion followed by inner membrane fusion. While the two fusion events are typically coupled in vivo, they can be decoupled in vitro due to different metabolic requirements for cofactors [25]. The complex process of mitochondrial fusion is orchestrated by a set of specific integral membrane proteins which belong to the highly conserved family of large dynamin-related GTPase proteins (DRPs). Among these, Mitofusin1/2 (MFN1/2) are members of the dynamin family and are embedded in the mitochondrial outer membrane, where they facilitate outer membrane fusion as large GTPases. On the other hand, Optic Atrophy 1 (OPA1), as another member of the dynamin family, is responsible for mediating inner membrane fusion. Importantly, any insufficiency or malfunction of these proteins can significantly impair mitochondrial fusion, thus leading to consequential defects in mitochondrial function [26]. In contrast, Dynamin-related protein 1 (DRP1), a large GTPase, is the key player in mitochondrial fission. Unlike mitofusins, DRP1 is located in the cytosol and requires association with several DRP1 receptors that are embedded in the outer mitochondrial membrane to be recruited to mitochondrial surface and facilitate fission. Mitochondrial fission factor (MFF), one important DRP1 receptor, plays a vital role in this process, as its depletion leads to significant mitochondrial elongation [27]. After being recruited to the mitochondrial surface, DRP1 assembles into oligomeric spirals that constrict the mitochondrial tubule. However, the final step of the scission process requires the involvement of another protein, Dynamin 2 (DYN2). Without DYN2, mitochondrial fission intermediates with narrow constrictions will accumulate, resulting in incomplete fission [28]. Although the machinery for mitochondrial fusion and fission are distinguishing, there is evidence of coordination between them, and both processes are linked to cellular metabolism [29].

### 3.3. Mitophagy and Mitochondrial Biogenesis

Mitophagy and mitochondrial biogenesis are essential cellular processes that regulate mitochondrial quality control and metabolic homeostasis in DMSCs. Mitophagy selectively eliminates damaged mitochondria, while mitochondrial biogenesis generates new ones. These processes are interdependent and regulated through mitochondrial fusion and fission, which help maintain an active mitochondrial network [30]. Mitochondrial fission is a crucial process for successful mitophagy as it facilitates the separation of damaged regions of the mitochondria from the active mitochondrial network during mitochondrial stress or injury. In addition, fission also contributes to mitochondrial biogenesis and quality control by generating new mitochondria. Conversely, mitochondrial fusion reduces mitophagy by diluting the impaired mitochondria. Therefore, maintaining a balance between mitochondrial biogenesis and mitophagy is crucial for the preservation of mitochondrial function and health in cells, and this balance is regulated by the opposing processes of mitochondrial dynamics.

Mitophagy is a highly selective degradation process that eliminates damaged and dysfunctional mitochondria and is conserved throughout evolution. The whole process is primarily conducted by PTEN-induced kinase 1 (PINK1), a serine/threonine kinase protein, and Parkin, a cytosolic E3-ubiquitin ligase. After mitochondrial membrane depolarization, PINK1 accumulates on the outer mitochondrial membrane (OMM) via the translocase of the outer membrane (TOM), followed by the recruitment of Parkin to mitochondria [31]. PINK1 initiates Parkin recruitment through phosphorylating ubiquitin at serine 65, which is typically attached to proteins on the OMM [32]. The phosphorylation of S65-phosphorylated ubiquitin (pS65-Ub) then facilitates the translocation of Parkin to the OMM, where it binds to pS65-Ub, allowing PINK1 to further phosphorylate serine 65 in Parkin’s ubiquitin-like domain (UBL) [32,33]. Interestingly, phosphorylated Parkin exhibits increased E3 ligase activity and a higher affinity for pS65-Ub, thereby promoting further recruitment and activation of Parkin and creating a positive feedback loop [33]. The coordinated activity of PINK1 and Parkin leads to the ubiquitination of mitochondria, which serves as a molecular signal for recruiting the autophagy machinery and promotes the engulfment of damaged mitochondria by autophagosomes which are ultimately delivered to the lysosome for degradation [34]. Furthermore, the receptor-mediated mitophagy pathway assumes a pivotal role, wherein an activated membrane receptor directly engages autophagosomal marker proteins (LC3/Atg8-like) for selective autophagy [35]. In current research, two distinct types of mitophagy receptors consistently localized to the OMM have been identified in mammalian cells, demonstrating the capability to recruit the autophagic machinery to mitochondria. This receptor repertoire comprises NIX/BNIP3L and BNIP3 within one family, while FUNDC1 represents the other mitophagy receptor [36,37]. Expanding beyond protein receptors, it is noteworthy that OMM-localized lipids also serve as highly effective baits for recruiting the mitophagy machinery [38].

Mitochondrial biogenesis is a tightly regulated process by which cells generate new fully functional mitochondria and increase overall mitochondrial mass. Although mitochondria are semiautonomous organelles that contain their own self-replicating genome, most mitochondrial proteins are nuclear-encoded. Therefore, a coordinated regulatory network of nuclear and mitochondrial genes is required for the generation of new mitochondria. The regulatory network underlying mitochondrial biogenesis is centered on peroxisome proliferator-activated receptor-gamma coactivator-1alpha (PGC-1α). This protein is commonly recognized as the primary regulator of this process. Following phosphorylation modification, PGC-1α translocates from the cytoplasm to the nucleus [39], where it triggers mitochondrial biogenesis through further coactivating the expression of nuclear respiratory factors-1/2 (NRF-1/2), estrogen-related receptor-α (ERR-α) and the mitochondrial transcription factor A (TFAM), the ultimate effector of mtDNA transcription and replication [40]. Studies have demonstrated that these transcription factors target a broad range of genes related to mitochondrial function, including the electron transport chain (ETC), detoxification response, protein import machinery, and mtDNA replication and transcription [40]. Notably, in addition to nuclear regulation which predominates mitochondrial biogenesis, mitochondria-related processes, such as protein import, mtDNA replication, transcription, and translation, also play a crucial role [40].

### 3.4. Mitochondria Transfer

Mitochondrial transfer, an emerging frontier in cellular regenerative medicine, has recently exhibited profound therapeutic potential. This phenomenon involves the intercellular transfer of mitochondrial DNA (mtDNA) from donor cells to recipient cells afflicted with dysfunctional mitochondria. In 2006, Spees et al. first observed this restorative capacity in mesenchymal stem cells (MSCs) through a co-culture system [41].

Mitochondrial transfer, a pivotal process for cellular rescue in the context of dysfunctional mitochondria, relies on intricate communication between donor and recipient cells. The transportation of vital cellular components and signals is facilitated by various structures, including tunnelling nanotubes (TNTs), gap junctions (GJs), extracellular vesicles (EVs), and cell fusion events (Figure 3). Among these, TNTs emerge as the most prominent mode of mitochondrial transfer, offering a glimpse into the fascinating world of intercellular communication.

The existence of TNTs, de novo structures connecting cells or complex cellular networks, was first reported by Rustom et al. in 2004 [42]. Serving as conduits for transporting cellular components and signals, TNTs facilitate the efficient exchange of mitochondria between cells [43]. Gap junction (GJ) proteins, known as connexins, play a crucial role in promoting the exchange of small molecules between neighboring cells. The intricate web of GJs enables the passage of mitochondria from donor to recipient cells, contributing to the overall cellular rescue process [44]. In addition to TNT-mediated transportation, mitochondria can also be encapsulated within microvesicles, typically ranging from 0.1 to 1 μm in diameter, and subsequently secreted outside of host cells [45]. The release and uptake of mitochondria through EVs offers an alternative mode of mitochondrial transfer, expanding the repertoire of communication pathways involved. Furthermore, mitochondrial transfer can occur either through partial cell fusion via TNT formation or complete cell fusion. While most studies in the field concentrate on the physiological release and uptake of mitochondria, some investigations have demonstrated that mitochondrial transfer can also be achieved through artificial means. The exploration of artificial approaches widens the scope of potential applications and challenges, encouraging researchers to explore novel avenues for therapeutic interventions [46].

### 3.5. Mitochondria and Damage-Associated Molecular Patterns

Damage-associated molecular patterns (DAMPs) serve as endogenous alarmins released from dying cells, eliciting recognition by pattern recognition receptors (PRRs) that subsequently instigate the innate immune system [47]. DAMPs originate from various cellular sources, including the nucleus, cytosol, plasma membrane, endoplasmic reticulum (ER), and mitochondria. Under normal physiological conditions, DAMPs are generally incapable of triggering PRR signaling due to the shielding effect of cellular membranes and compartmentalization within the cell. Nevertheless, unintended cell injury or death may induce significant alterations in the permeability of different cellular compartments, enabling DAMPs to gain physical access to PRRs and initiating inflammatory responses [48]. Recent studies have identified mitochondria as a critical source of DAMPs, with numerous mitochondrial constituents and metabolic products capable of triggering inflammation upon release into the cytosol or extracellular milieu (Figure 4) [48]. Mitochondrial DAMPs (MtDAMPs) exhibit a minimum of two evolutionarily conserved molecular imprints inherited from their bacterial ancestry: N-formyl peptides (NFPs) and mtDNA. MtDNA and N-formyl peptides inadvertently liberated from compromised mitochondria and cells are distinctly acknowledged by PRRs within the innate immune system, Toll-like receptor (TLR) 9 and formyl peptide receptor (FPR), respectively [49,50]. The implications of these discoveries are profound, as multiple independent research teams have unveiled diverse signal transduction cascades activated in response to mitochondrial dysfunction, leading to inflammatory reactions [43].

To gain a comprehensive understanding of the intricate interplay between mitochondria and the innate immune system requires a journey into the realm of the endosymbiotic theory of mitochondria. Eons ago, the fusion of α-proteobacteria, precursors to contemporary Gram-negative bacteria, with either archaeal or eukaryotic hosts gave rise to a novel cellular entity [51]. Over epochs of evolution, this amalgamation transformed into the mitochondria we recognize today, accompanied by the loss or integration of a substantial portion of the proteobacterium’s DNA into the nuclear genome [51]. Central to this narrative is the retention of unmethylated CpG motifs within the mitochondrial genome, a hallmark absent in the human genome but ubiquitous in bacterial DNA [52]. This dichotomy in CpG motif presence paved the way for the evolution of PRRs within the innate immune system, designed to identify unmethylated CpG motifs as pathogen-associated molecular patterns (PAMPs) [52]. However, the innate immune system, in its innate simplicity, struggles to distinguish between unmethylated CpG motifs of bacterial origin and those originating from mitochondria. Under normal physiological conditions, this is not problematic as mtDNA is confined to mitochondria. In the context of mitochondrial stress, cellular damage, or necrosis, mtDNA can enter the cytoplasm or extracellular space, exerting an immunostimulatory effect, albeit as a DAMP rather than a PAMP [53,54]. Containing the inflammatogenic unmethylated CpG DNA motifs, mtDNA serves as a ligand for TLR9, a constituent of the highly conserved family of PRRs. Internalization of DNA containing motifs stimulates endolysosomal TLR-9, setting in motion a complex proinflammatory response. Additionally, intracellular signaling through cyclic GMP-AMP synthase (cGAS) and stimulator of interferon response cGAMP interactor 1 (STING1) on the ER has been identified as one such cascade, activated by mtDNA [55]. Further exploration into the relationship between mitochondrial dysfunction and inflammation reveals the involvement of the inflammasome. Studies have shown that mtDNA and reactive oxygen species (ROS) act as inducers of the inflammasome, offering an additional pathway through which mitochondrial dysfunction leads to inflammatory responses [54].

Another subset of DAMPs, rooted in the functional legacy of mitochondria’s bacterial ancestry, manifests as NFPs. Analogous to their bacterial counterparts, mitochondria necessitate NFPs for translation initiation [56]. Typically, these endogenous NFPs remain sequestered within the confines of mitochondria. However, under conditions of cellular injury or heightened mitochondrial stress, akin to mtDNA, NFPs exhibit the capacity to be released into the cytosol or extracellular space. Upon such release, NFPs actively engage with critical receptors such as FPR, FPR-like 1, or FPRL-like 2 situated on phagocytic leukocytes, thereby orchestrating an immune response [56].

## 4. Mitochondria and Multi-Directional Differentiation of DMSCs

### 4.1. Mitochondria and Osteoblastic Differentiation of DMSCs

The intricate relationship between mitochondrial function, energy production, and osteoblastic differentiation of DMSCs has recently emerged as a compelling area of investigation. Osteoblastic differentiation of DMSCs is a complex and energy-demanding process, highlighting the significance of energy production. Mitochondrial OXPHOS, a key process involved in cellular ATP production, has been found to have multifaceted roles in DMSCs, extending beyond energy metabolism. However, the relationship between OXPHOS and glycolysis in this process remains a controversial issue.

In a recent study, researchers have demonstrated that indicators measuring mitochondrial activity, specifically OXPHOS, directly enhanced by increased TFAM expression, were positively correlated with enhanced osteoblastic differentiation of SHED [57]. Furthermore, another study has elucidated the negative impact of reduced mitochondrial activity and ATP levels on the osteoblastic differentiation ability of SHED isolated from patients with Leigh syndrome, a rare neurological disorder caused by mitochondrial dysfunction [58]. In addition, it was shown that aerobic respiratory function in the mitochondria and the intracellular ATP levels were decreased by melatonin at physiological concentrations, leading to inhibited osteoblastic differentiation of hPDLSCs [59]. However, melatonin treatment at pharmacological concentrations can promote the osteoblastic differentiation of hPDLCs by accelerating cellular energy supply through mitochondrial OXPHOS [60].

Interestingly, enhanced mitochondrial activity during osteoblastic differentiation may serve a purpose other than higher energy demand. For instance, induction of pseudohypoxia has been shown to shift energy metabolism from OXPHOS to glycolysis while concurrently enhancing the proliferation and osteoblastic differentiation of hPDLCs [61]. However, the interconversion of mitochondrial OXPHOS and glycolysis during osteoblastic differentiation has been a topic of intense debate among researchers. The complexity of this issue has led to significant differences in opinion, with even the same research group reporting contradictory experimental results. Previous studies on MSCs had suggested that only OXPHOS, and not glycolysis, was upregulated during osteoblastic differentiation [62]. In contrast, a recent in vitro experiment conducted on DFSCs demonstrated the induction of some glycolysis markers during osteoblastic differentiation, indicating that glycolytic energy production is also not dispensable [63]. In an in vivo study, researchers discovered that during osteoblastic differentiation of DPSCs, there is a reduction in mitochondrial OXPHOS, along with an increase in glycolysis activity [64]. Interestingly, when ferutinin, a natural compound, was added to initiate the osteoblastic differentiation of DPSCs, a decrease in glycolytic activity was observed [65]. Collectively, these studies demonstrate that the regulation of mitochondrial activity and energy production is crucial for osteoblastic differentiation, although the specific roles of glycolysis and OXPHOS remain controversial. Further research is required to fully comprehend the mechanisms underlying the metabolic shift and its role in the differentiation of DMSCs.

In addition, it is significant to note that mitochondria OXPHOS is not only involved in cellular ATP production but also in the production of ROS [66]. Although ROS can cause cell dysfunction and tissue death, only an unregulated level of the ROS is hazardous, as the physiological upregulation of ROS is vital for the MSC self-renewal and ROS levels can influence the direction of MSCs differentiation [24,67]. In our previous research, we observed a reduction in ROS levels in hPDLSCs after osteoblastic induction, which is in line with some previous studies [66,68]. Moreover, recent findings have revealed that decreased mitochondria-related ROS levels and enhanced mitochondria OXPHOS are associated with the promotion of the osteoblastic differentiation of i-PDLSCs (inflammatory periodontal ligament stem cells) by gallic acid [69]. In contrast, the loss of membrane potential of mitochondria significantly reduces the level of osteoblastic differentiation of hPDLSCs [69]. Additionally, induced mitochondrial ROS accumulation was shown to suppress the osteoblastic differentiation of hPDLSCs, which can be reversed by FoxO1 or curcumin [70,71]. Therefore, it is evident that the regulation of mitochondrial function and ROS production plays a crucial role in the osteoblastic differentiation of DMSCs and could be a promising avenue for developing new strategies to promote tissue regeneration.

The biological activities of mitochondria in the MQC system are also implicated in the osteoblastic differentiation of DMSCs. Specifically, mitochondrial autophagy and biogenesis play crucial roles. Interestingly, despite appearing as opposite processes, both mitochondrial autophagy and biogenesis have been shown to promote osteoblastic differentiation of DMSCs. In vitro research has demonstrated that mitochondrial biogenesis and network formation can be enhanced through the BZF-PGC-1α pathway without affecting mitochondrial membrane potential (MMP), resulting in improved osteoblastic differentiation of LS-SHED [72]. Furthermore, a previous study has shown that mitophagy, the specific organelle autophagy process, can promote osteoblastic differentiation of hDPSCs through the BMP/Smad pathway triggered by the accumulation of amorphous calcium phosphate (ACP) [73]. Additionally, the participation of the ubiquitin ligase Smurf1 in mediating mitophagy has been demonstrated [74]. Notably, the induction of PINK1/Parkin-mediated mitophagy was observed during osteoblastic differentiation of DPSCs, as confirmed by analysis of mitochondrial structure through BioTEM [64]. Moreover, in vivo experiments have further supported these findings by demonstrating that the activation of PINK1/Parkin-mediated mitophagy, regulated by exosomes, can improve the osteoblastic differentiation of PDLSCs [75]. Interestingly, iPDLSCs isolated from periodontitis patients exhibit compromised osteogenesis relative to hPDLSCs obtained from healthy donors, suggesting that decreased osteoblastic differentiation of PDLSCs is the most critical cause of periodontitis [76]. Further research has indicated that UCHL1 downregulates PINK1/Parkin-mediated mitophagy, leading to the suppression of osteoblastic differentiation in PDLSCs during periodontitis-associated inflammation by inhibiting the BMP2/Smad signaling pathway [77]. In addition, mitophagy assists in eliminating dysfunctional mitochondria and lowering ROS levels in developing cells exposed to mild stress, thereby ensuring cell survival. Thus, restoring the osteoblastic differentiation potential of iPDLSCs can be achieved through the improvement of mitophagy, as demonstrated in a rat model of periodontal inflammation [78]. Additionally, osteoblastic differentiation of hPDLCs can be promoted by melatonin at pharmacological concentrations, via improving mitochondrial fusion and inhibiting mitochondrial fission [60]. In a recent study, researchers delved into the phenomenon of mitochondrial transfer and its potential impact on the osteoblastic differentiation capacity of PDLSCs cultured under low stiffness conditions. The study’s findings shed light on the restorative role of mitochondrial transfer, providing evidence of its ability to partially rescue the osteoblastic differentiation capacity of PDLSCs [79].

In summary, the regulation of mitochondrial activity and energy production, as well as the quality control system of mitochondria, play critical roles in the osteoblastic differentiation of DMSCs (Table 1). However, the specific roles of OXPHOS and glycolysis in this process remain controversial and require further research. Understanding the mechanisms underlying these processes may lead to the development of novel strategies for promoting tissue regeneration. In the future, it will be essential to investigate the roles of other mitochondrial pathways and their interactions with other cellular processes in osteoblastic differentiation. Advances in this field have the potential to lead to the development of innovative therapies for various bone disorders.

### 4.2. Mitochondria and Odontoblastic Differentiation of DMSCs

The intricate relationship between mitochondrial function and odontoblastic differentiation of dental DMSCs has captured the attention of researchers in recent years. While it is widely accepted that optimal mitochondrial function is necessary for the successful differentiation of DMSCs into odontoblasts, the exact mechanisms behind the reprogramming of mitochondrial OXPHOS and glycolysis during DMSC differentiation into odontoblasts remain a contentious topic of debate. Conflicting results have been reported in various studies, underscoring the need for further investigation into the role of mitochondria in this process.

Undifferentiated MSCs are considered to depend on anaerobic glycolysis to supply most of the energy required for cellular functions since they are always isolated from the hypoxic niches [80]. In addition, studies have reported that increased glycolysis rates, in conjunction with decreased OXPHOS, are essential for MSCs to evade ROS-induced oxidative damage and supply the necessary substrates for their proliferation [81]. However, as MSCs differentiate, their energy metabolism pathway shifts from glycolysis to mitochondrial oxidative metabolism [82], indicating that mitochondria can modulate MSCs differentiation via bioenergy conversion. Accumulating evidence has indicated that a transition in energy production from glycolysis to aerobic metabolism, accompanied by an increase in mitochondrial respiratory functions, is a critical step in the successful differentiation of MSCs [83]. For instance, a recent study demonstrated that mitochondrial nanoprobes facilitate odontoblastic differentiation of DPSCs by increasing MMP to induce more ATP synthesis, which suggests that increased MMP intensity could serve as a driving force for the differentiation of DPSCs into odontoblasts [84]. Consistent with this finding, our previous research suggested that mitochondrial respiratory function was enhanced during odontoblastic differentiation of DPCs, derived from the ectomesenchyme during tooth development [66]. And the increased mitochondrial function was a prerequisite for odontoblastic differentiation since rotenone, an inhibition of OXPHOS, can impede that process [66]. In addition, the promotion of odontoblastic differentiation in DPCs involves the modulation of mitochondrial respiratory function and mitochondrial ROS homeostasis by SIRT4 [85]. Furthermore, in vivo experiments showed that vital processes of mitochondrial energy metabolism were impaired when gene expression associated with this function was blocked in DMSCs derived from Sirt6 gene knockout mice, leading to a diminished capacity for differentiation [86]. However, another in vitro study indicated that both mitochondrial OXPHOS and glycolysis were enhanced during the initial phase of hDPSCs differentiation into odontoblast, which is in contrast to previous conclusions in the differentiation of MSCs [87]. Notably, the increase in glycolysis and mitochondrial OXPHOS activation was found to be independent of oxygen environments, indicating that the characteristics of glycometabolism are directly related to the odontoblastic differentiation of hDPSCs [87]. Researchers suggest that the increase in glycolysis serves as a protective mechanism for the cells, preventing the excessive generation of ROS resulting from increased mitochondrial OXPHOS and oxidative damage to cellular components [88]. While numerous observations have demonstrated that elevated levels of ROS stimulate differentiation in various cell types, excessive and uncontrolled levels of ROS can induce cellular damage [89,90,91]. Intriguingly, our previous study observed a decrease in ROS levels during odontoblastic differentiation of DPCs, which may relate to the increasing intracellular NADH level, an effective antioxidant enhanced by mitochondrial respiratory function [66]. These results suggest that the differentiation of DMSCs into odontoblasts is not a straightforward process from glycolysis to mitochondrial OXPHOS but rather a complex dynamic one. Nevertheless, these studies collectively emphasize the crucial importance of mitochondrial function in the differentiation of DMSCs into odontoblasts and highlight the need for a deeper understanding of the potential molecular mechanisms behind the interplay between mitochondrial metabolism and cellular differentiation.

The role of the MQC system in the differentiation of DMSCs into odontoblasts has been gradually elucidated. Recent studies have provided profound insights into the critical role that the MQC system plays in regulating the differentiation of DMSCs into odontoblasts.

It has been demonstrated that the transition of mitochondrial dynamics toward fusion and inhibition of mitochondrial autophagy can enhance the odontoblastic differentiation of DMSCs. A recent study on odontogenesis demonstrated that mitochondrial fusion accelerated the differentiation into odontoblasts and dentin formation of dental papilla cells, which was further enhanced by inhibiting DRP1, a major mitochondrial division factor [92]. However, a recent study showed that a reduction in mitochondrial fission and an imbalance of mitochondrial dynamics in HDPCs induced by LPS limited mineralization and odontoblastic differentiation [93]. Furthermore, mitophagy has also been shown to play a crucial role in driving odontoblastic differentiation in DMSCs. An investigation indicated that hypoxia-induced mitophagy in HDPCs was triggered by the phosphorylation of FUNDC1, a vital molecule situated on the outer mitochondrial membrane. Silencing FUNDC1 expression was observed to deactivate hypoxia-induced mitophagy, as demonstrated by decreased protein expression levels of LC3II [94]. This impediment in mitophagy was found to compromise odontoblastic differentiation, underscoring the pivotal role of it in mediating the hypoxia-driven enhancement of odontoblastic differentiation in HDPCs [94]. Another previous investigation into pediatric hyperbilirubinemia has proposed that oxidative stress, characterized by an increase in ROS and the activation of mitochondrial apoptosis, induced apoptosis in SHED to prevent differentiation into dentin [95].

Taken together, the regulation of mitochondrial function and quality control plays a crucial role in the odontoblastic differentiation of DMSCs (Table 2). Further research into the complex interactions between mitochondria and odontoblastic differentiation could help us better understand the nature of cellular differentiation and the mechanisms of related diseases, leading to promising therapeutic targets for pulp repair and regeneration.

### 4.3. Mitochondria and Neurogenic Differentiation of DMSCs

The significance of mitochondria in the neurogenesis of DMSCs is becoming increasingly apparent. A mounting body of evidence reveals that enhanced mitochondrial respiration and a moderate level of ROS are essential for the neurogenic differentiation of DMSCs.

According to Kato et al., the researchers observed that during the neurogenic differentiation of SHED, the mitochondrial respiratory function increased, while the overall mass of mitochondria remained unchanged [96]. These findings were further reinforced by experiments in which inhibitors of the mitochondrial respiratory chain and mitochondrial uncouplers were found to inhibit the neurogenic differentiation of SHED [96]. Additionally, the authors discovered that to maintain stem cell properties before differentiation, the activity of mitochondria was reduced [96]. In a recent study, it has been demonstrated that both mitochondrial respiration and ROS production could be stimulated by the activation of transient receptor potential canonical cation channel type 1 (TRPC1) through pulsed electromagnetic fields, resulting in a synergistic effect on DPSC neurogenesis when combined with graphene [97]. Whereas another study has highlighted excessive ROS production triggered by mitochondrial Ca2+ accumulation and overload led to impairments in neurite development during the differentiation of SHED into dopaminergic neurons [98]. Thus, the delicate balance between ROS levels and mitochondrial respiration must be carefully considered, as high levels of ROS can be potentially harmful to cellular development, while moderate levels have been demonstrated to prime mitochondrial adaptations that enhance neurogenic survival and differentiation [99] via a process known as mitochondrial hormesis or mitohormesis [100].

Furthermore, the differentiation of DMSCs into neurons involves intricate regulation of mitochondrial activities which includes mitochondria dynamics, mitophagy, and mitochondrial biogenesis, et cetera. Studies have demonstrated that during this process, the dynamic balance between mitochondrial biogenesis and mitophagy shifts towards the former. To shed light on the mechanisms behind the differentiation of human periodontal ligament stem cells (hPDLSCs) into neuronal cells induced by moringin, RNA-seq analysis, and the Reactome database reveal that most genes involved in the mitophagy pathway and oxidative stress are downregulated in hPDLSCs pretreated with moringin (hPDLSCs-MOR) [101]. For instance, both the genes orchestrating phagophore formation (MAP1LC3B, GABARAP, GABARAPL1/2) and the pivotal initiator of the process (PINK1) were observed to be downregulated [101]. Additionally, through the downregulation of the genes that participate in mitochondrial fusion (e.g., MFN1) and upregulation of those in fission (e.g., DNM1L), the appropriate equilibrium between mitophagy and mitochondrial biogenesis is maintained in hPDLSCs-MOR, making the target genes promising candidates for supporting neurological therapies [101]. Interestingly, increased ROS levels also have a strong correlation with mitochondrial biogenesis, which can either be positively or negatively regulated, depending on the type of induced stress [102]. Acute or mild stress has the potential to stimulate the expression of PGC-1α and enhance mitochondrial biogenesis as a result of the engagement of mitochondrial quality control mechanisms while severe or long-lasting stress can result in the opposite effects [102,103]. Sun et al. also indicated that MFF insufficiency induced the overproduction of mitochondrial ROS and subsequently decreased mitochondrial biogenesis mediated by the downregulation of PGC-1α, which resulted in defects in the neurogenic differentiation of SHED [98]. Whereas, by promoting mitochondrial biogenesis and accelerating ROS scavenging, the administration of folic acid has been shown to restore neurite development in SHED differentiated into dopaminergic neurons due to MFF insufficiency [98].

In summary, extensive research has been conducted in recent years regarding the function of mitochondria in the neurogenesis of DMSCs (Table 3). Researchers highlight the significance of mitochondrial biogenesis and fine-tuning mitochondrial function for successful neurogenic differentiation. Thus, mitochondria in the neurogenesis of DMSCs hold promising potential for supporting neurological therapies and developing new therapeutic strategies for neurodegenerative diseases.

### 4.4. Mitochondria and Angiogenic Differentiation of DMSCs

Angiogenic differentiation is a fundamental process in regenerative endodontic procedures (REPs), enabling the replacement of damaged tissues and successful pulpal repair after injury and inflammation [104]. The switch from mitochondrial OXPHOS to glycolysis has been shown to be essential for the angiogenic differentiation of DMSCs. Mitochondria, as the primary source of cellular energy, play a vital role in facilitating this metabolic shift. The switch to glycolysis reduces the demand for oxygen and enables the delivery of maximum oxygen concentration to tissues perfused by the blood vessels. This metabolic shift is particularly significant in neovascularization, where endothelial cells rely on glycolysis as their primary bioenergetic process, despite the availability of nearby oxygen sources [105]. Moreover, a recent study has confirmed that the induction of genes, which regulates the metabolic transition from OXPHOS to glucose oxidation mediated by hypoxia-inducible factor-1α (HIF-1α), is critical in mediating the paracrine angiogenic effects of SHED under hypoxic conditions [106]. In vivo, the angiogenic capacity of SHED is impaired by the silencing or inhibiting of HIF-1α, underscoring the significance of mitochondria in regulating the proper stem cell differentiation into functional endothelial cells [106]. Furthermore, the role of mitochondria in regulating cellular oxidative stress and ROS is critical in maintaining cellular health and preventing oxidative damage, ensuring proper cell growth, survival, and regulating the signaling pathways that drive the differentiation process. ROS, primarily generated by mitochondria, regulate the proliferation and angiogenesis of DMSCs under normal oxygen levels [106]. Notably, hPDLSCs that have differentiated into endothelial cells appear to be more vulnerable to lipopolysaccharide (LPS), resulting in the production of higher levels of ROS than their undifferentiated counterparts [107] (Table 4). 

## 5. Strategies, Gaps and Directions in Mitochondria-Mediated Regulatory Mechanisms and Therapy in DMSC Differentiation

Stem cell-based tissue engineering stands at the forefront of innovative approaches for the restoration of damaged, injured, or absent tissues, wherein stem cells undergo differentiation into specific phenotypes, creating a regenerative microenvironment. The utilization of MSCs in mediating regeneration for applications in tissue engineering and cytotherapy has witnessed a discernible surge. Initially scrutinized as instrumental entities in regenerative medicine aimed at tissue replacement, the efficacy of MSC-based regeneration, despite its considerable potential, remains inconsistently realized. In clinical trials, administered MSCs seldom exhibited differentiation and successful engraftment into host tissues, despite demonstrating positive effects in various disease models [108]. Given the pivotal role of mitochondria in modulating MSC functions, strategies targeting mitochondria hold promise in optimizing MSC-based regenerative therapy. Notably, mitochondria contribute significantly to stem cell pluripotency and lineage specification through metabolic and MQC system modulations. For instance, MSCs directly exhibit antioxidant properties by scavenging free radicals, augmenting mitochondrial functions and donating mitochondria [44,109,110]. Thus, reagents capable of modulating mitochondrial metabolism, and pharmaceuticals known as antioxidants/ROS scavengers for oxidative suppression increasingly show efficacy to counteract cell aging and pathologies via modulating stem cell specification [110,111,112]. Furthermore, progress in inhibitors and activators of mitochondrial fission and fusion opens avenues for modulating mitochondrial dynamics across various stem cell models [80,113]. However, current pharmacological therapeutics encounter challenges in precisely modulating mitochondrial function. Of particular note are recent strides in gene editing technologies and nanoparticle-based drug delivery systems targeting mitochondria, such as mitochondrial-targeted transcription activator-like effector nucleases (mitoTALEN) and mitochondrially targeted zinc-finger nucleases (mtZFN), in conjunction with nanoparticles for mitochondria-targeted drug delivery, collectively offering substantial therapeutic potential [114,115,116].

Mitochondrial dysfunction has been implicated in a diverse array of diseases across the medical spectrum due to its central role in the organismal homeostasis. Recent studies have highlighted the potential of MSCs to address mitochondrial defects and compensate for malfunction through intercellular mitochondrial transfer [117,118,119]. This emerging field holds promise for developing therapeutic strategies to treat diseases associated with mitochondrial dysfunction. Mitochondrial donation by MSCs presents a faster and more cost-effective approach to replace dysfunctional mitochondria in diseased cells or tissues, compared to mitochondrial biogenesis, offering an efficient means to alleviate disease conditions [120]. Additionally, the use of isolated and artificial mitochondria as a future therapy shows considerable potential with isolated mitochondria becoming a readily available therapy in regenerative medicine [121]. Nevertheless, further investigations are required to determine the optimal dosage, packaging, delivery methods, and ethical considerations associated with using isolated mitochondria. Furthermore, the integration of advanced imaging techniques and omics technologies has yielded valuable insights into the dynamic changes occurring within mitochondria during the differentiation of MSCs [122]. Systemic approaches, such as live detection on mitochondrial biogenesis and dynamics, in-depth profiling of mitochondrial metabolic and biochemical status, and rigorous mapping of mtDNA genome and mitochondrial protein heterogeneity, will enable the comprehensive evaluation of mitochondrial changes under different conditions.

Despite significant advancements in our understanding of mitochondria-mediated regulatory mechanisms in the differentiation of MSCs, several crucial aspects in this field still require thorough investigation. Firstly, the precise molecular mechanisms underlying the lineage-specific differentiation of MSCs mediated by mitochondria remain largely elusive. Elucidating these intricate regulatory pathways is essential for comprehending the differentiation potential of MSCs and developing targeted therapeutic strategies. Secondly, the crosstalk between mitochondria and other cellular signaling pathways involved in MSCs fate determination demands further exploration. Unraveling the intricate interplay between mitochondria and other organelles will provide valuable insights into the overall regulatory network governing stem cell biology. Additionally, the impact of mitochondrial dysfunction and aging on MSC differentiation represents an ongoing area of investigation, as their influence on the regenerative capacity of MSCs has yet to be fully understood [123]. Moreover, there is a critical need for the identification of mitochondrial signatures associated with health and disease, along with the development of mitochondria-targeted interventions in translational studies. Addressing these knowledge gaps will significantly advance our understanding of mitochondria-mediated regulatory mechanisms and pave the way for effective clinical interventions in various pathological conditions. Furthermore, precise manipulation of mitochondria in a cell type- and spatiotemporal-specific manner, as well as the optimization of MSC donor selection, dosage, and the development of efficient cell delivery techniques, pose unexplored challenges for medical interventions targeted at treating pathological conditions associated with mitochondrial dysfunction.

As introduced in the beginning, due to the numerous advantages of DMSCs over traditional MSCs, they have emerged as a highly promising cell source. Modulating mitochondrial function and improving mitochondrial quality, along with exploring mitochondria-related agents, offer effective strategies for controlling and activating DMSC functions, enabling their utilization in regenerative medicine and the treatment of age-related diseases. Additionally, integrating biological research on novel molecular targets associated with mitochondrial signatures in diseases with the development of advanced tools for precise modulation holds significant potential for optimizing stem cell-based mitochondrial therapeutics and promoting regeneration in the context of organismal aging and pathologies. For instance, in future regenerative medicine, a promising strategy involves the regulation of ROS generation through various approaches, aiming to enhance the therapeutic efficacy of DMSCs and improve the prognosis of patients with terminal diseases. Furthermore, exploring the potential of mitochondria-based therapeutic strategies, such as mitochondrial transplantation or manipulation of mitochondrial metabolism, shows promise for augmenting the differentiation of mesenchymal stem cells and advancing regenerative therapies (Figure 5). Overall, the comprehensive understanding and precise manipulation of mitochondrial activities in a cell-specific and spatiotemporal manner, along with optimization of DMSC-based interventions, present unexplored challenges that, when addressed, will significantly contribute to the success of regenerative medicine in the context of aging and various pathologies.

## 6. Conclusions and Future Perspectives

DMSCs are currently considered ideal seed cells for tissue regeneration through tissue engineering methods due to their capability to self-renew and differentiate into various cell lineages, as well as their accessibility through minimally invasive surgery without ethical concerns. This paper reviews the functions and mechanisms of mitochondria in the multi-directional differentiation of DMPCs, highlighting the crucial roles that mitochondrial energy metabolism, mitochondrial biogenesis, mitochondrial autophagy, and mitochondrial dynamics play in the multi-directional differentiation of DMSCs (Figure 6). Therefore, gaining insight into the roles of mitochondria in controlling the multi-directional differentiation of DMSCs is significant. The regulation of mitochondrial function, methods to control mitochondrial quality, and the exploration of mitochondria-related drugs that may regulate and activate DMSCs have the potential to accelerate the application of cell therapy in clinical practice, making it a promising strategy in regenerative medicine. In addition, mitochondrial transfer is a highly efficient and cost-effective physiological process that can replace dysfunctional mitochondria in diseased cells/tissue, making it a promising strategy for attenuating a wide range of disease conditions when compared to mitochondrial biogenesis. Further, the process of mitochondrial transfer represents a sophisticated interplay of regulatory structures. Understanding the underlying communication pathways and regulatory elements may provide valuable insights into potential therapeutic strategies for mitochondrial disorders. The diverse means by which mitochondria can be transferred, including through EVs and artificial approaches, hint at the vast potential of this field for advancing cellular regenerative medicine. As research progresses, unraveling the complexities of mitochondrial transfer holds promise for unlocking new frontiers in cellular therapies and disease treatment. The recognition of mitochondria as critical sources of DAMPs and inducers of inflammatory signaling pathways represents a significant advancement in our understanding of cellular immune responses. The intricate interplay between DAMPs and inflammation is a rapidly evolving area of research, with broad implications for the understanding and treatment of a range of pathological conditions. Understanding the specific molecular pathways involved in this process may facilitate the development of novel therapeutic strategies aimed at combating inflammation and the diseases associated with it. Whereas, despite the progress that has been made, understanding of the role of mitochondria in the differentiation of DMSCs remains in its early stages. It is yet to be determined whether the regulation of mitochondria is the determining factor in the differentiation direction of DMSCs, the mechanisms of mitochondrial energy metabolism and the signaling targets that regulate DMSCs differentiation remain unclear, the molecular basis of the link between mitochondrial dynamics and cell differentiation is yet to be fully understood, and the regulation of mitophagy and biogenesis in DMSCs differentiation is still not elucidated. Moreover, current studies on mitochondrial regulation of multi-directional differentiation of DMSCs have focused on osteoblastic differentiation, odontoblastic differentiation, and neurogenic differentiation. Although DMSCs can differentiate into adipocytes and myocytes, there are few reports on their mitochondrial regulation, there is still a lack of reports on mitochondrial regulation in these histiocytes. All these factors hinder the improvement of the function of DMSCs and their application in regenerative medicine.

Overall, although the field of DMSC differentiation is rapidly evolving, interest in the contribution of mitochondria has considerably increased. However, much remains to be discovered about the full extent of mitochondria’s potential role and the mechanisms by which they exert their influence.

## Figures and Tables

**Figure 1 biomolecules-14-00012-f001:**
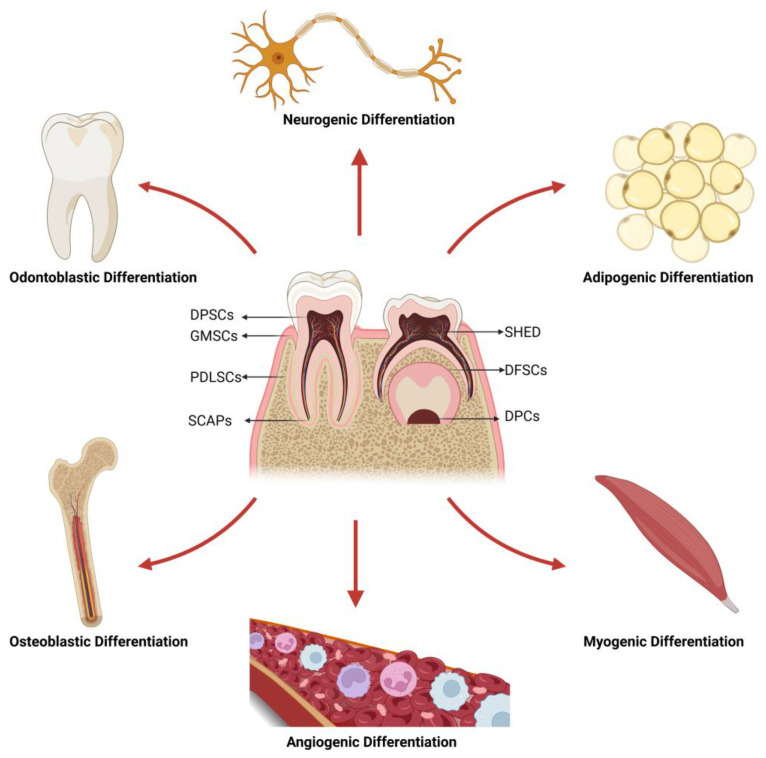
The location and multi-directional differentiation potential of DMSCs. DMSCs can be isolated from various tissues in the oral cavity and have multi-directional differentiation ability. Abbreviations: GMSCs, gingival mesenchymal stem cells; DPSCs, dental pulp stem cells; SHED, stem cells from human exfoliated deciduous teeth; PDLSCs, periodontal ligament stem cells; SCAPs, stem cells from apical papilla; DFSCs, dental follicle stem cells.; DPC, dental papilla cells. Figure was created using BioRender.com.

**Figure 2 biomolecules-14-00012-f002:**
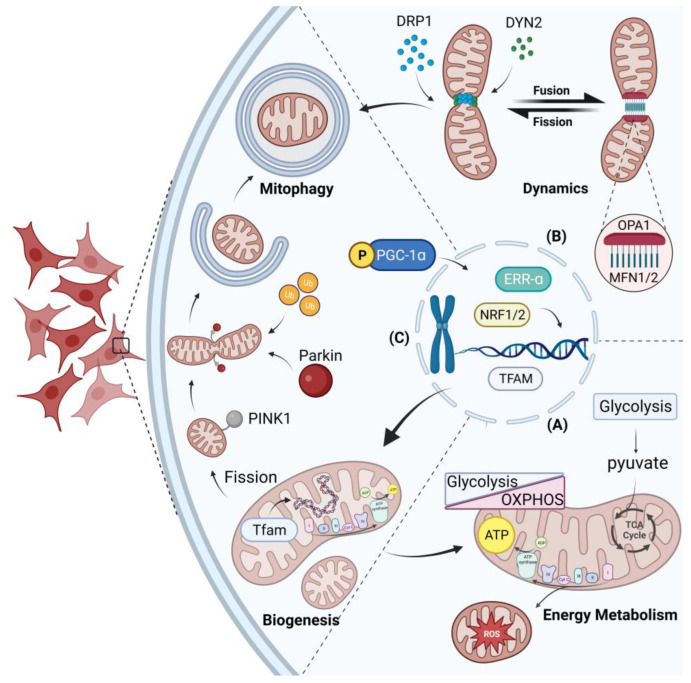
Mitochondrial control system and mitochondrial energy metabolism. The mitochondrial quality control system regulates mitochondrial biogenesis, dynamics, and mitophagy. Energy metabolism is the primary function of mitochondria in DMSCs. (**A**) Mitochondrial energy metabolism: Mitochondrial energy metabolism is a complex process that involves converting pyruvate, which is produced by glycolysis into ATP through the TCA cycle in the mitochondrial matrix. These energetic molecules are then utilized by ETC and OXPHOS to produce ATP. In addition, an increased OXPHOS metabolic state is associated with higher production of ROS by the mitochondria; (**B**) Mitochondrial dynamics: Mitochondrial dynamics is a continuous cycle of fusion and fission that maintains an active mitochondrial network until the onset of mitophagy. Three essential GTPases—MFN1, MFN2, and OPA1—control mitochondrial fusion, with MFN1 and MFN2 acting on the OMM and OPA1 acting on the IMM. Conversely, mitochondrial fission involves the coordination of two GTPases, namely DRP1 and DYN2, to separate damaged mitochondrial components from the functional network. DRP1 causes spiral constriction around the OMM in a GTP-dependent manner, which is subsequently followed by the recruitment of DYN2 to the OMM in order to facilitate membrane division; (**C**) Mitochondrial biogenesis and Mitophagy: Mitophagy is a type of selective autophagy that targets dysfunctional mitochondria from DMSCs by recruiting them to mitochondrial phagosomes, which are subsequently degraded by lysosomes. Accumulation of PINK1 on the outer mitochondrial membrane (OMM) occurs via TOM, which then recruits Parkin through PINK1-dependent ubiquitin phosphorylation. This induces the formation of mitochondrial phagosomes, ultimately triggering mitophagy. Mitochondrial biogenesis is a complex regulatory mechanism involving several transcription factors, including PGC1α, NRF1, NRF2, ERRα, and TFAM, that work in concert to generate new mitochondria. During periods of physiological stress, PGC1α is phosphorylated, allowing it to translocate to the nucleus and coactivate downstream transcription factors. Upon activation, NRFs and ERRα enhance the transcription of genes related to ETC, while TFAM regulates replication and transcription of mtDNA after translocating to mitochondria. In addition, PGC1α-mediated mitochondrial biogenesis supports OXPHOS levels in DMSCs, ensuring sufficient energy production to meet metabolic demands. Abbreviations: TCA, the tricarboxylic acid; ROS, reactive oxygen species; MFN1/2, Mitofusin1/2; Opa1, optic atrophy factor 1; DRP1, dynamin-related protein 1; DYN2, Dynamin 2; TFAM, mitochondrial transcription factor A; PGC-1α, peroxisome proliferator-activated receptor γ-coactivator 1α; NRF1/2, nuclear respiratory factor 1/2; PINK1, PTEN-induced kinase 1; ub, ubiquitin. Figure was created using BioRender.com.

**Figure 3 biomolecules-14-00012-f003:**
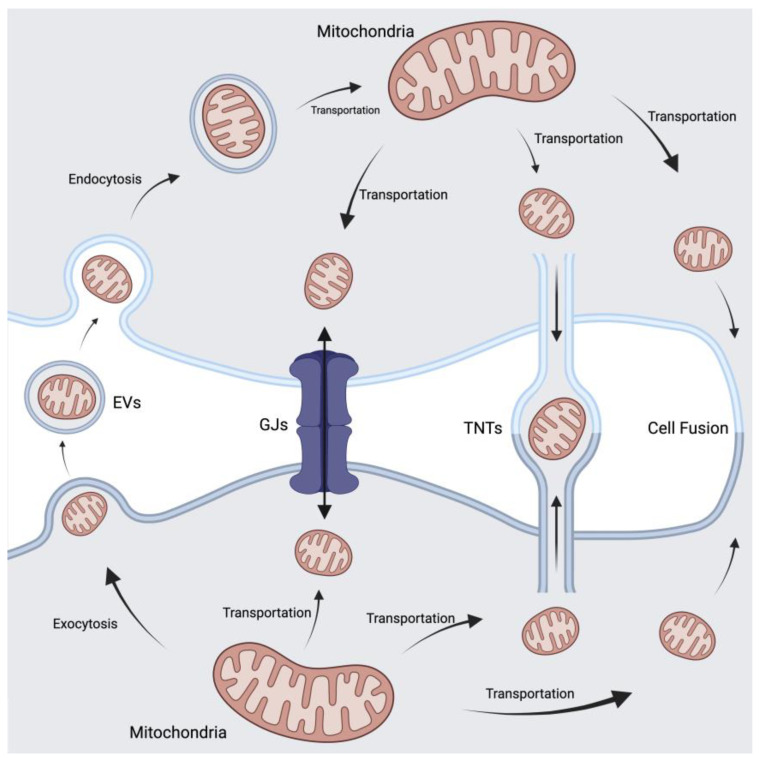
Mechanisms of mitochondria transfer. The intercellular transfer mechanisms of mitochondria can be classified into four distinct categories: extracellular vesicle-mediated ejection for targeted delivery, intercellular transportation via gap junctions, formation of tunnelling nanotubes facilitating mitochondrial movement between cells, and redistribution of mitochondria achieved by complete cell fusion. Abbreviations: EVs, extracellular vesicles; GJs, gap junctions; TNTs, tunnelling nanotubes. Figure was created using BioRender.com.

**Figure 4 biomolecules-14-00012-f004:**
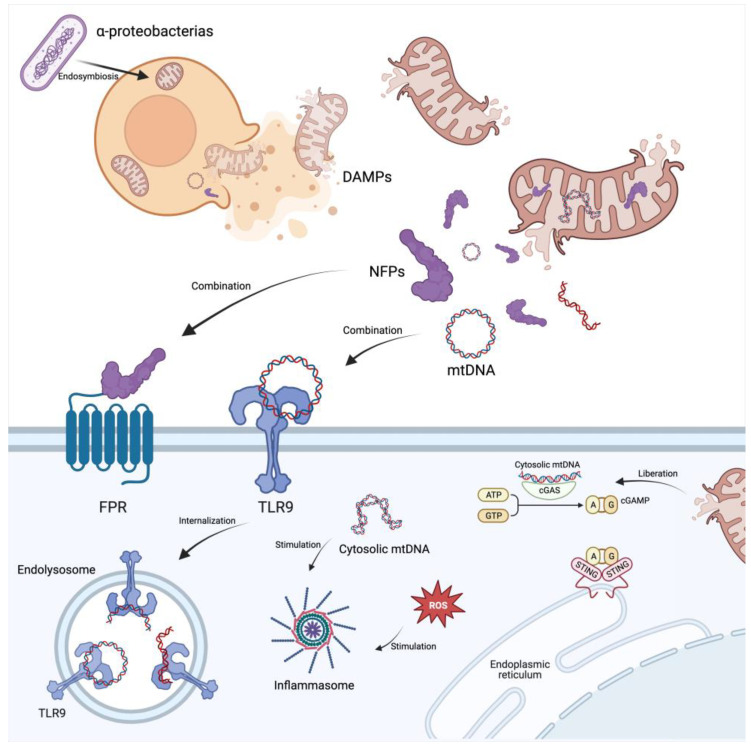
Mitochondria and damage-associated molecular patterns. Mitochondrial DAMPs are potent immunological activators due to the bacterial ancestry. Cellular injury or death results in the release of DAMPs, including mitochondrial constituents, into the extracellular milieu. Mitochondrial DAMPs such as NFPs and mtDNA gain access to PRRs in the innate immune system. TLR9 recognizes mtDNA’s unmethylated CpG motifs, activating a proinflammatory reaction. Simultaneously, NFPs interact with receptors like FPR, orchestrating an immune response. Furthermore, mitochondrial dysfunction triggers the cGAS–STING pathway and inflammasome activation, through cytosolic mtDNA and ROS, offering supplementary routes for inflammation. Abbreviations: DAMPs, damage-associated molecular patterns; NFPs, N-formyl peptides; mtDNA, mitochondria DNA; PRRs, pattern recognition receptors; FPR, formyl peptide receptor; TLR9, Toll-like receptor 9; cGAMP, cyclic GMP-AMP; STING1, stimulator of interferon response cGAMP interactor 1. Figure was created using BioRender.com.

**Figure 5 biomolecules-14-00012-f005:**
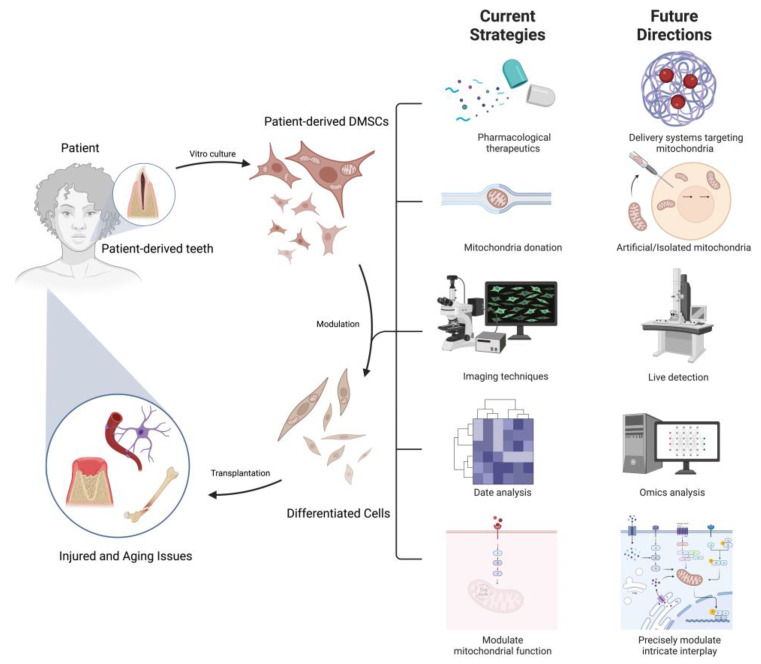
Current strategies and future directions in mitochondria-mediated regulatory mechanisms and therapy in DMSCs differentiation. The mitochondria-mediated regulatory mechanisms and therapy in DMSC differentiation represents a promising avenue in regenerative medicine. This approach involves extracting DMSCs from patients, followed by the nuanced modulation of their mitochondrial function and activity, thereby steering their differentiation trajectories. Ultimately, these manipulated DMSCs find application in tissues and organs impacted by damage and aging. Current key strategies predominantly encompass pharmacological interventions, utilizing MSCs for intercellular mitochondrial transfer, imaging techniques, data analysis and modulating mitochondrial function. Promising future directions include the mitochondrial-targeted drug delivery systems, the investigation of artificial/isolated mitochondria as a regenerative therapy, the integration of advanced imaging and omics technologies to comprehensively evaluate mitochondrial changes and precise manipulation of mitochondria in a cell-specific and spatiotemporal manner. The comprehensive understanding of mitochondrial activities in DMSCs, optimization of DMSC-based interventions, and exploration of mitochondria-based therapeutic strategies, such as mitochondrial transplantation, are key challenges for advancing regenerative medicine in the context of aging and pathologies. Figure was created using BioRender.com.

**Figure 6 biomolecules-14-00012-f006:**
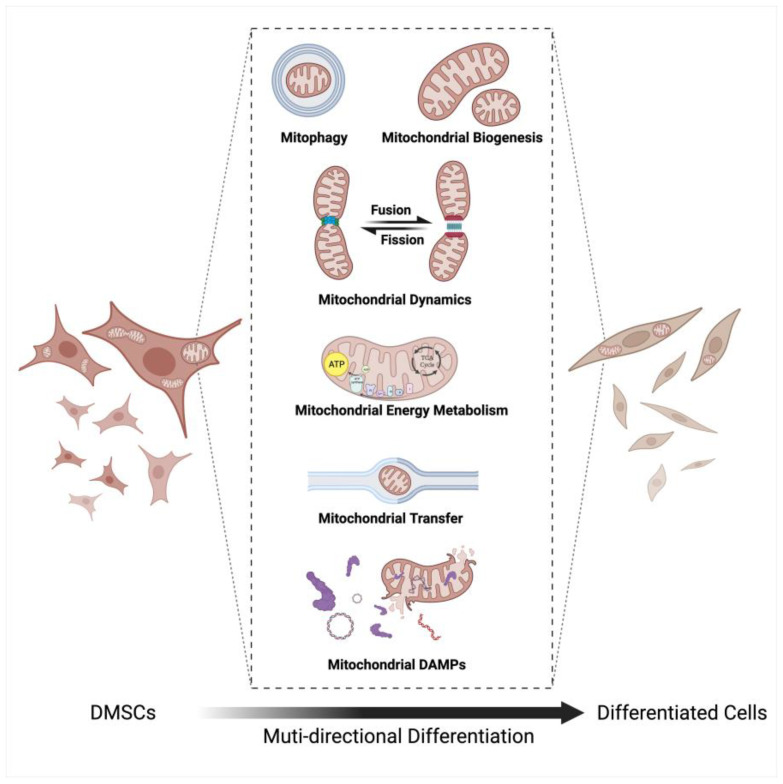
Mitochondria and multi-directional differentiation of dental-derived mesenchymal stem cells. Mitochondrial energy metabolism, mitochondrial biogenesis, mitophagy, mitochondrial dynamics, mitochondria transfer and mitochondrial DAMPs play crucial roles in the multi-directional differentiation of DMSCs. Abbreviations: DMSCs, dental-derived mesenchymal stem cells. Figure was created using BioRender.com.

**Table 1 biomolecules-14-00012-t001:** Effects of mitochondrial activity on the osteoblastic differentiation of DMSCs under different conditions.

DMSC Types	Condition	Mitochondrial Activity	Relationship with Differentiation	References
** *Mitochondrial energy metabolism* **				
SHED	Increased TFAM expression	Increase OXPHOS	Promote differentiation	[57]
SHED	Patient with Leigh syndrome	Inhibit OXPHOS	Inhibit differentiation	[58]
DPSCs	In vivo experiment	Increase glycolysis and inhibit OXPHOS	Accompany differentiation	[64]
DPSCs	Ferutinin	Increase glycolysis	Initiate differentiation	[65]
DFSCs	In vitro experiment	Increase glycolysis	Accompany differentiation	[63]
hPDLSCs	Melatonin at physiological concentrations	Inhibit OXPHOS	Inhibition differentiation	[59]
hPDLSCs	Melatonin at pharmacological concentrations	Increase OXPHOS	Promote differentiation	[60]
hPDLCs	Pseudohypoxia	The shift from OXPHOS to glycolysis	Promote differentiation	[61]
hPDLCs	Osteoblastic induction	Decreased ROS	Accompany differentiation	[66]
hPDLSCS	Decreased MMP	Decrease OXPHOS	Inhibit differentiation	[69]
hPDLSCs	Fox1/curcumin	Reduce accumulated ROS	Restore the differentiation potential	[70,71]
i-hPDLCs	Gallic Acid	Decrease ROS and increase OXPHOS	Promote differentiation	[69]
** *Mitochondrial quality control system* **				
SHED	BZF-PGC-1ɑ pathway	Improve mitochondrial biogenesis	Promote differentiation	[72]
DPSC	Observation by BioTEM	Presentation of Mitophagy	Accompany differentiation	[64]
hDPSCs	Induction of mitophagy	Improve Mitophagy	Promote differentiation	[73]
PDLSCs	Exosomes	Activation of mitophagy	Promote differentiation	[75]
PDLSCs	UCHL1	Inhibit mitophagy	Inhibit differentiation	[78]
iPDLSCs	Rat model of periodontal inflammation	Improve mitophagy	Restore the differentiation potential	[60]
hPDLCs	Melatonin at pharmacological concentrations	Improve mitochondrial fusion and inhibit mitochondrial fission	Promote differentiation	[79]
hPDLCs	Low stiffness culture	Mitochondria transfer	Restore the differentiation	[67]

**Table 2 biomolecules-14-00012-t002:** Effects of mitochondrial activity on the odontoblastic differentiation of DMSCs under different conditions.

DMSCs Type	Condition	Mitochondrial Activity	Relationship with Differentiation	References
** *Mitochondrial energy metabolism* **				
**DPSCs**	Mitochondrial nanoprobes	Increase OXPHOS	Promote differentiation	[84]
**hDPSCs**	The initial phase of differentiation	Increase Glycolysis and OXPHOS	Promote differentiation	[87]
**DPCs**	Odontogenic induction	Increase OXPHOS	Promote differentiation	[66]
**DPCs**	Rotenone	Inhibit OXPHOS	Inhibit differentiation	[66]
**DPCs**	Increase NADH level	Decrease ROS	Promote differentiation	[66]
**DPCs**	Melatonin-mediated malic enzyme 2	Increase OXPHOS	Promote differentiation	[1]
**DPCs**	SIRT4	Increase OXPHOS and decrease ROS	Promote differentiation	[85]
**DMSCs**	Sirt6 gene knockout mice	Inhibit OXPHOS	Inhibit differentiation	[86]
** *Mitochondrial quality control system* **				
**DPCs**	Inhibition of DRP1	Improve mitochondrial fusion	Promote differentiation	[92]
**DPCs**	Melatonin-mediated malic enzyme 2	Improve mitochondrial fusion	Promote differentiation	[1]
**HDPCs**	LPS	Reduce mitochondrial fission	Inhibit differentiation	[93]
**HDPCs**	Hypoxia-induced phosphorylation of FUNDC1	Induce mitophagy	Promote differentiation	[94]
**HDPCs**	Silencing FUNDC1	Inhibit mitophagy	Inhibit differentiation	[95]

**Table 3 biomolecules-14-00012-t003:** Effects of mitochondrial activity on the neurogenic differentiation of DMSCs under different conditions.

DMSCs Type	Condition	Mitochondrial Activity	Relationship with Differentiation	References
** *Mitochondrial energy metabolism* **				
**SHED**	Inhibitors of ETC and mitochondrial uncouplers	Inhibit OXPHOS	Inhibit differentiation	[96]
**SHED**	mitochondrial Ca2+ overload	Produce excessive ROS	Inhibit differentiation	[98]
**SHED**	MFF insufficiency	Produce excessive ROS	Inhibit differentiation	[98]
**DPSCs**	TRPC1	Increase OXPHOS and ROS	Promote differentiation	[97]
** *Mitochondrial quality control system* **				
**SHED**	Folic acid	Promote mitochondrial biogenesis and accelerate ROS scavenging	Restore the differentiation potential	[98]
**SHED**	Downregulation of PGC-1α	Mitochondrial biogenesis	Inhibit differentiation	[98]
**hDPSCs**	Downregulation of genes involved in fusion	Inhibit mitochondrial fusion	Promote differentiation	[101]
**hDPSCs**	Upregulation of genes involved in fission	Promote fission	Promote differentiation	[101]
**hPDLSCs**	Moringin	Inhibit mitophagy and oxidative stress	Inhibit differentiation	[89]

**Table 4 biomolecules-14-00012-t004:** Effects of mitochondrial activity on the angiogenic differentiation of DMSCs under different conditions.

DMSCs Type	Condition	Mitochondrial Activity	Relationship with Differentiation	References
**SHED**	HIF-1α	Increase glycolysis and inhibit OXPHOS	Promote	[106]
**SHED**	Inhibit HIF-1α	Increase ROS	Inhibit	[106]
**hPDLSCs**	LPS	Increase ROS	More pronounced after differentiation	[107]

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
