# Peer review of "Mitochondria in Multi-Directional Differentiation of Dental-Derived Mesenchymal Stem Cells"

_biomolecules, 2023, doi:10.3390/biom14010012_

Round 1

Reviewer 1 Report

Comments and Suggestions for Authors

This is an exciting and thoroughly written paper. I do not have anything to add or edit. The figures are of good quality and the organization is good. Lines 92-94 need correction.

Comments on the Quality of English Language

The English language and writing is of adequate quality.

Author Response

1. Summary

Thank you very much for taking the time to review this manuscript. Please find the detailed responses below and the corresponding corrections highlighted changes in the re-submitted files.

3. Point-by-point response to Comments and Suggestions for Authors

Comments 1: This is an exciting and thoroughly written paper. I do not have anything to add or edit. The figures are of good quality and the organization is good. Lines 92-94 need correction.

Response 1: Thanks for your careful reading and positive comments. We apologize for the ignorance in lines 92-94. Sentences unrelated to this review in lines 92-94 of the initial manuscript has been deleted.

3. Response to Comments on the Quality of English Language

Point 1: The English language and writing is of adequate quality.

Response 1: We appreciate your positive assessment of the English language. We hope our work will provide high-quality material that meets the needs and expectations of our readers.

Reviewer 2 Report

Comments and Suggestions for Authors

The manuscript showed that the review aims to overview the regulatory mechanisms in multi-directional differentiation of DMSCs mediated by mitochondrial energy metabolism and the MQC system, further indicating the potential therapeutic implications of modulating mitochondrial function to enhance DMSC-based regenerative medicine applications.

It is an excellent review regarding understanding the mitochondrial energy metabolism and the MQC system in Multi-Directional Differentiation of Dental Derived Mesenchymal Stem Cells. However, there are a few suggestions to improve it as follows.

Comment;

1. The authors explain that dental-derived mesenchymal stem cells (DMSCs) include GMSCs, DPSCs, SHEDs, PDLSCs, DFSCs, and DPCs. However, the functional roles of mitochondria and mitophagy are different during the differentiation process of each cell. Therefore, it would be good to help readers understand the mitochondria and mitophagy signaling pathways during the differentiation process for each cell. 

2. The authors explain cell damage occurs when ROS accumulates during DMSC differentiation. However, there is no explanation for the elimination of ROS during these differentiation processes. Therefore, elucidation of the ROS removal mechanism during the differentiation of DMSCs into various cells is necessary.

Reviewer 3 Report

Comments and Suggestions for Authors

This paper discussed the significance of dental-derived mesenchymal stem cells (DMSCs) in tissue regeneration via regulating mitochondria energy metabolism and the MQC system. In addition, the author also discussed the impact of modulating mitochondrial function and their potential therapeutic implications for enhancing DMSC-based regenerative medicine applications. I have the following comments.

1.       Table 1: "The terms 'vivo' and 'vitro' should be corrected to 'in vivo' and 'in vitro.' Please also specify 'LS' in the context of 'Patient with LS.'

2.       Please include a figure summarizing current strategies and future directions for utilizing mitochondria-based therapy in tissue regeneration would enhance the clarity of the manuscript.

3.       Please consider incorporating images adapted from other papers, especially when discussing phenomena such as mitochondria transfer.

4.       The information provided by the author on Mitochondria and damage-associated molecular patterns lacks sufficient detail. I suggest the author elaborate a bit more.

5.       It would be helpful to include a table to summarize “Mitochondria and angiogenic differentiation of DMSCs”.

6.       In figure3, Mitochondria transfer and Mitochondria and damage-associated molecular patterns are missing. Please ensure its inclusion.

Round 2

Reviewer 3 Report

Comments and Suggestions for Authors

N/A